# Acute Oxidative Stress Can Paradoxically Suppress Human NRF2 Protein Synthesis by Inhibiting Global Protein Translation

**DOI:** 10.3390/antiox12091735

**Published:** 2023-09-07

**Authors:** Kaitlin M. Pensabene, Joseph LaMorte, Amanda E. Allender, Janessa Wehr, Prabhjot Kaur, Matthew Savage, Aimee L. Eggler

**Affiliations:** Department of Chemistry, Villanova University, Villanova, PA 19085, USA

**Keywords:** NRF2, hydrogen peroxide, protein synthesis, antioxidant response, acute oxidative stress, mechanisms of NRF2 regulation

## Abstract

The NRF2 transcription factor is a master regulator of the cellular oxidant/electrophile response and a drug target for the prevention/treatment of chronic diseases. A major mechanism of NRF2 activation is its escape from rapid degradation, and newly synthesized NRF2 induces cytoprotective protein expression through its cognate antioxidant response elements (AREs). However, oxidative stress can also inhibit global protein translation, thereby potentially inhibiting NRF2 protein accumulation. H_2_O_2_ has been shown to be a relatively weak inducer of NRF2 in comparison with electrophiles. In the current study, we evaluated whether levels of H_2_O_2_ that activate the NRF2/ARE pathway inhibit NRF2 protein synthesis in HaCaT keratinocytes. A weak maximum induction was observed for H_2_O_2_ in comparison with electrophiles, both for NRF2 protein accumulation and ARE reporter activation (~10-fold compared to ≥100-fold activation). At similar H_2_O_2_ concentrations, both NRF2 protein synthesis and global protein synthesis were inhibited. The manganese porphyrin antioxidant MnTMPyP rescued both global protein synthesis and NRF2 protein synthesis from H_2_O_2_ inhibition and increased ARE reporter activation. Similar results were observed for the diphenol di-*tert*-butylhydroquinone (dtBHQ). In conclusion, induction of the NRF2/ARE pathway by H_2_O_2_ and dtBHQ-derived oxidative species can be limited by inhibition of NRF2 protein synthesis, likely by arrest of global protein synthesis.

## 1. Introduction

Protein translation is a tightly regulated, energy-demanding process that is modulated by conditions of cell stress [1,2,3]. In general, stressors inhibit global protein synthesis, conserving resources and resolving damage through the synthesis of a specific subset of proteins. In yeast, H_2_O_2_ increases targeted translation of antioxidant proteins, while limiting global protein synthesis [4,5]. A variety of mechanisms in yeast and humans have been identified by which oxidative stress both inhibits global protein synthesis and increases targeted protein synthesis [3,6,7]. In humans, the NRF2 transcription factor is a master regulator of the response to oxidative stress, inducing the transcription of cytoprotective proteins such as glutathione synthesis enzymes [8], and as such, is a drug target for prevention and treatment of a wide range of chronic diseases [9,10,11]. Many natural product molecules that induce a cellular antioxidant response act through NRF2 [12]. H_2_O_2_ is commonly used in the NRF2 field as a representative oxidative insult in cell culture, with the induction of the pathway generally observed at 100 to 600 µM H_2_O_2_ (e.g., [13,14,15]). Several mechanisms have been identified by which the synthesis of the NRF2 protein itself can be increased in response to H_2_O_2_ at concentrations in this range, including internal ribosomal entry sites (IRESs) and a G-quadruplex structure in the 5′ untranslated region of NRF2 mRNA [16,17,18] and recruitment of NRF2 mRNA to active polysomes [16]. However, it has not yet been investigated whether global protein synthesis inhibition might outcompete these mechanisms and limit NRF2 protein synthesis in response to levels of H_2_O_2_ that activate NRF2-driven gene expression, or other reactive oxygen species (ROS).

The question of whether H_2_O_2_/oxidative stress increases or decreases NRF2 protein synthesis has significant implications for NRF2-driven gene expression, given the mechanism of NRF2 activation. Under basal conditions, NRF2 is constitutively synthesized and targeted for ubiquitination by the KEAP1–cullin 3-E2 complex, followed by degradation by the 26S proteasome, resulting in a very short half-life of ~20 min for the NRF2 protein [19]. Human KEAP1 contains 27 cysteines, and modification of specific cysteines by H_2_O_2_, soft electrophiles, and other cysteine-reactive molecules [20] halts NRF2 ubiquitination and degradation [21]. For example, H_2_O_2_ targets a cluster of four KEAP1 cysteines, C226/613/622/624 [14]. Upon escaping KEAP1 repression, NRF2 accumulates and translocates into the nucleus, binding to copies of its cognate antioxidant response element (ARE) and driving gene expression. When the KEAP1 complex is rendered inactive, protein synthesis is required for NRF2 to accumulate in the cell [21]. Brusatol, a small molecule anti-cancer agent that inhibits global protein translation, was first identified as an NRF2 inhibitor, illustrating the strong dependence of NRF2 accumulation on global protein synthesis [22]. Thus, although H_2_O_2_/oxidative stress activates the pathway, that activation would be limited if NRF2 protein synthesis were inhibited by the stalling of global protein synthesis by the oxidative stress.

Interestingly, H_2_O_2_ has been reported to be a fairly weak activator of the NRF2/ARE pathway compared to other NRF2 activators in side-by-side comparisons [14,23]. In mouse embryonic fibroblasts (MEFs), modest stabilization of NRF2 protein required 130 µM H_2_O_2_, and the maximum amount of NRF2 stabilized, at 400 µM H_2_O_2_, was less than that obtained for various other NRF2 activators [14]. These included the electrophile 2-cyano-3,12-dioxoolean-1,9-dien-28-oic acid (CDDO)-imidazolide (at 10 nM), the electrophilic signaling molecule 15-deoxy-Δ^12,14^-prostaglandin J_2_ (at 10 µM), ebselen (at 10 µM), which forms seleno-sulfide adducts with target protein cysteine residues, and compound 16, a non-electrophilic, competitive protein–protein interaction inhibitor of NRF2 and KEAP1 binding (at 10 µM). In mouse keratinocytes, both in cells and in whole mice, electrophiles (sulforaphane, *tert*-butylhydroquinone, and 15-deoxy-Δ12,14-prostaglandin J2) avidly activated the NRF2 pathway, as expected; however H_2_O_2_ or UV exposure, which caused cellular oxidative stress, did not [23]. Thus, at least in certain cell types such as embryonic fibroblasts and keratinocytes, some mechanism limits the induction of the NRF2/ARE pathway by H_2_O_2_, as compared to its induction by electrophiles. One explanation may be the suppression of NRF2 and global protein translation by H_2_O_2_.

Previously, we observed opposing stimulatory and inhibitory effects of H_2_O_2_/ROS on the induction of the NRF2/ARE pathway by an electrophile in human keratinocytes. The electrophilic phytochemical sulforaphane was combined with the ROS-generating small molecule di-*tert*-butylhydroquinone (dtBHQ). Sulforaphane, which targets KEAP1 C151 [14,24,25], is the subject of nearly one hundred clinical trials that span a wide range of chronic diseases [26]. The combination synergistically activated ARE-driven gene expression [27,28], showing that H_2_O_2_/ROS can boost ARE-driven gene expression. However, dtBHQ unexpectedly significantly inhibited NRF2 protein accumulation at the same concentrations that enhanced ARE-driven gene expression [28]. In addition, H_2_O_2_ suppressed the level of NRF2 protein accumulation ± sulforaphane.

Thus, in this work we asked whether concentrations of H_2_O_2_ that activate the NRF2/ARE pathway also suppress NRF2 protein synthesis, in HaCaT keratinocytes. These non-tumorigenic cells are spontaneously immortalized, with mutations in p53 and an otherwise stable chromosome content [29,30], and have been used to elucidate NRF2/ARE regulation mechanisms [31]. In addition, we investigate whether H_2_O_2_ and dtBHQ block NRF2 protein synthesis by stalling global translational machinery. This is the first study to assess the net effect of ROS on endogenous NRF2 protein synthesis at concentrations of oxidative species that activate NRF2. We find that concentrations of H_2_O_2_ that activate ARE reporter expression inhibit both NRF2 protein synthesis and global protein synthesis. An antioxidant, the superoxide dismutase mimetic MnTMPyP, rescues both global protein synthesis and NRF2 protein synthesis from H_2_O_2_ inhibition and increases ARE reporter activation. Similar results were observed for the redox-cycling oxidizable diphenol di-*tert*-butylhydroquinone (dtBHQ). Ultimately, these findings indicate that inhibition of global protein synthesis by acute oxidative stress can stall NRF2 synthesis, thereby limiting the NRF2 response.

## 2. Materials and Methods

### 2.1. Chemicals and Reagents

Unless stated otherwise, all chemicals were purchased from Sigma-Aldrich. D,L-sulforaphane (Toronto Research Chemicals Inc. #S699115, North York, ON, Canada) was dissolved in anhydrous DMSO (≥99.9%) to 50 mM. Single-use aliquots were stored at −80 °C. Cycloheximide (#C7698) and dtBHQ (#112976) were prepared daily in DMSO from powder. Stock H_2_O_2_ (9.8 M; #216763) was diluted for each experiment in phosphate-buffered saline (PBS). Manganese (III) tetrakis(1-methyl-4-pyridyl)porphyrin pentachloride (MnTMPyP; A.G. Scientific #M3307) was dissolved in PBS to 4 mM, with aliquots stored in the dark at 4 °C. MG-132 (Selleck Chemicals, #S2619, Houston, TX, USA) was prepared as a 10 mM solution in DMSO and stored as single-use aliquots at −80 °C. Cell culture reagents and transfection reagents were purchased from Life Technologies, except L-glutamine and fetal bovine serum (FBS; Atlanta Biologicals, Flowery Branch, GA, USA). The Problock Gold protease inhibitor cocktail was purchased from Gold Biotechnology. Fast Green (#F7252) was dissolved in 1% (*v*/*v*) acetic acid to a concentration of 0.1% (*w*/*v*). DNase I Recombinant (#04536282001) was dissolved in 50% glycerol containing 20 mM HEPES and 1 mM MgCl_2_ and stored at 4 °C. The reagents in the Click-&-Go Plus 647 OPP Protein Synthesis Assay Kit (Click Chemistry Tools, #1496, Scottsdale, AZ, USA) were stored at 4 °C, except the AZDye 647 Azide Plus, which was dissolved in DMSO and stored in the dark at −20 °C per manufacturer instructions. Primary anti-NRF2 antibody is from D1Z9C from Cell Signaling Technology (#12721), and peroxidase AffiniPure goat anti-rabbit IgG was purchased from Jackson ImmunoResearch Laboratories (West Grove, PA, USA).

### 2.2. Cell Culture

The HaCaT cell line (Cell Lines Service; L#300493-4212, Heidelberg, Germany) was maintained at 37 °C in 5% CO_2_ in phenol red-free and sodium-pyruvate-free high-glucose Dulbecco’s modified Eagle’s medium (DMEM) supplemented with 10% fetal bovine serum, 15 mM HEPES (4-(2-hydroxyethyl)-1-piper-azineethanesulfonic acid) pH 7.2, and 4 mM L-glutamine (referred to as complete media). Complete media was stored in the dark at 4 °C, with aliquots heated to 37 °C just prior to use. Cells were maintained between 50 and 80% confluency during both propagation and experiments, passaging every three days. The following treatments were at these final concentrations: 12.5 μM MnTMPyP, 10 μM MG-132, and 50 mg/mL cycloheximide.

### 2.3. ARE Reporter Assay

HaCaTs were seeded at 5.0 × 10^4^ cells/mL in 24-well plates and transfected 18 h later using 1 µL/well TransIT-2020 (Mirus Bio, Madison, WI, USA) and Opti-MEM (Life Technologies, Carlsbad, CA, USA) to deliver 45 μg/well pGL4.37 Luc ARE plasmid (Promega, Madison, WI, USA) and 15 μg/well Renilla plasmid (Promega) in 50 µL/well Opti-MEM reduced serum medium. After 4 h of transfection, media was replaced with 1 mL complete media. Cells were given 18–24 h recovery time after transfection. Media was replaced with 2 mL fresh media immediately before adding treatments directly to the wells. Total DMSO was equal across all treatments and did not exceed 0.1% (*v*/*v*). After 18 h, cells were rinsed with PBS and then lysed with Passive Lysis Buffer (Promega) and a freeze–thaw cycle. After transferring lysates to a 96-well plate, ARE reporter activity (firefly luciferase) was first assessed by adding 25 µL D-Luciferin (Biotium, Fremont, CA, USA) to each well, mixing, and measuring luminescence with a ClarioStar BMG Labtech luminometer. Renilla reporter activity was then assessed in the same manner using 25 µL of Aquaphile Coelenterazine (Biotium). Relative units of reporter activation were calculated as the firefly luciferase (ARE-driven) values divided by the Renilla luciferase values. Data were normalized to vehicle treatment alone.

### 2.4. Western Blot

Cells were seeded in 35 mm dishes at 2.80 × 10^5^ cells/dish in 2 mL of complete media and were left to recover overnight. Sulforaphane, dtBHQ, H_2_O_2_, and MnTMPyP treatments were added directly to the wells immediately following replacement with 3 mL complete media. Total DMSO was equal across all treatments and did not exceed 0.1% (*v*/*v*). When designing plate layouts for experiments, care was taken to avoid any neighboring well effect of the treatments with dtBHQ, as previously reported for *tert-*butyl hydroquinone (tBHQ) [32], by placing dishes apart from each other in the incubator. In NRF2 synthesis experiments, where indicated, MG-132 was added to the wells 10 min post-treatment, and translation elongation inhibitor cycloheximide was added with sulforaphane. After the 2 h treatment period, media was aspirated, and cells were rinsed with PBS, then lysed with 100 µL of lysis buffer (Cellytic M with 1% (*v*/*v*) protease inhibitor cocktail). Freshly made 10% Tris-glycine polyacrylamide gels and polyvinylidene difluoride (PVDF) membranes were used for electrophoresis and Western blotting. Blots were blocked in 5% dry milk in Tris-buffered saline with 1% Tween 20. Both the anti-NRF2 primary and anti-rabbit secondary antibodies were diluted 1:1000. We showed previously that this NRF2 primary antibody specifically detects NRF2 in multiple cell types as a doublet of bands at ~100 kDa on 10% Tris-glycine SDS PAGE gels [33]. Blots were probed using a SnapID apparatus (Millipore, Burlington, MA, USA) as per manufacturer’s instructions. The c-Digit imaging system (LI-COR) was used to visualize the detected bands after incubation with West Dura Substrate (Thermo Fisher, Waltham, MA, USA). Membranes from Western blotting analysis were stained using Fast Green solution, then washed in distilled water and dried before imaging. Images were obtained using the EC3 Imaging System (UVP) with ethidium bromide emission.

### 2.5. Global Protein Synthesis Assay

Cells were seeded and treated according to the Western blot protocol. Following an 80 min treatment, O-propargyl-puromycin OPP (2 μM final) or DMSO vehicle were added to the wells. After a 40 min incubation, cells were washed with PBS and lysed in 100 µL of 50 mM Tris-HCl with 1% SDS and 1% (*v*/*v*) protease inhibitor cocktail. Lysates were digested with DNase I and 2.5 mM MgSO_4_ before staining with the Click-&-Go Plus 647 OPP Protein Synthesis Assay Kit according to the manufacturer’s protocol. Samples were kept in the dark, precipitated with 20% TCA, and washed four times with acetone. Freshly made 10% Tris-glycine polyacrylamide gels were used for electrophoresis. Gels were imaged in the Cy5 channel on a Typhoon FLA 9500 imager. Gels were subsequently stained with Coomassie dye for total protein, rinsed with MilliQ water, and re-imaged in the Cy5 channel on the Typhoon. Global protein synthesis was quantitated by determining the intensities of the total protein synthesis signal and the total protein signal in ImageJ [34] and dividing the former by the latter. Data were normalized to the +OPP condition data.

### 2.6. Statistical Analysis

Data are presented as mean ± one standard deviation. Outliers were detected using the interquartile range (IQR) test, in which data points that were below (quartile 1 − (1.5 × IQR)) or above (quartile 3 + (1.5 × IQR)) were excluded. Statistical significance was analyzed utilizing a Student’s unpaired, two-tailed t-test for comparison of two conditions. Results were considered statistically significant at a *p*-value < 0.05.

## 3. Results

### 3.1. H_2_O_2_ Is a Relatively Poor NRF2/ARE Activator in Human Keratinocytes

Given that H_2_O_2_ relatively weakly activates the NRF2/ARE pathway in other studies, including in mouse keratinocytes [14,23], we compared the activation of an ARE reporter in human keratinocytes by H_2_O_2_ and various electrophiles. In HaCaTs, sulforaphane increases the activity of an ARE reporter to a much greater extent relative to vehicle only (~120-fold maximum, at 20 μM) than H_2_O_2_ (~10-fold maximum, at 150 μM) ((Figure 1A). For comparison, two other electrophilic NRF2 activators, bardoxolone methyl (CDDO-Me) and dimethyl fumarate, were tested in the same system. These also have much higher fold-induction than H_2_O_2_ in the same assay (360 ± 50-fold for bardoxolone methyl, at 2 µM, and 98 ± 9-fold for dimethyl fumarate at 100 µM, Appendix A).

Accordingly, as shown in Figure 1B, treatment with a broad range of H_2_O_2_ concentrations results in significantly less NRF2 protein accumulation compared with 2.5 µM sulforaphane, which activates the ARE reporter by 7.3 ± 0.8-fold. Thus, the accumulation of NRF2 protein by H_2_O_2_ is limited by some mechanism in these keratinocyte cells compared to sulforaphane. As expected, the ribosome inhibitor cycloheximide completely blocks sulforaphane-induced NRF2 protein accumulation, confirming that nascent NRF2 protein synthesis (in addition to KEAP1 cysteine modification) is necessary for NRF2 protein accumulation in HaCaT cells. Interestingly, higher concentrations of H_2_O_2_ (e.g., 200 µM) suppress the total amount of NRF2 protein compared to lower concentrations of H_2_O_2_, while total protein levels remain constant. This result is consistent with the hypothesis that H_2_O_2_ can inhibit nascent NRF2 protein synthesis.

### 3.2. H_2_O_2_ Inhibits NRF2 Protein Synthesis

To test whether H_2_O_2_ inhibits NRF2 protein synthesis, cells were treated with the proteasome inhibitor MG-132. Due to its short half-life of ~20 min [35], NRF2 protein rapidly accumulates when its degradation is blocked, allowing NRF2 protein synthesis activity to be measured by the extent of NRF2 accumulation (Figure 2A, first two lanes). NRF2 protein synthesis is inhibited by H_2_O_2_ in a dose-dependent manner with near-complete inhibition at 100 µM H_2_O_2_ (Figure 2A). This concentration correlates with H_2_O_2_ levels required to increase ARE reporter activity 5–10 fold (Figure 2B). Thus, H_2_O_2_ concentrations that effectively release NRF2 from KEAP1 inhibition and/or act on downstream targets likely also inhibit NRF2 protein synthesis.

To determine whether an antioxidant would both rescue NRF2 protein synthesis and increase ARE reporter activation, cells were co-treated with H_2_O_2_ and MnTMPyP, a manganese porphyrin superoxide dismutase mimetic (Figure 2A). MnTMPyP neutralizes several classes of reactive species, as it can also decompose the highly oxidizing species peroxynitrite [36]. It has a low level of catalase activity, with a rate several logs lower than that of its superoxide dismutase activity. It can also reduce production of the highly oxidizing hydroxyl radical, by reducing superoxide concentrations that would otherwise react with sufficiently high concentrations of H_2_O_2_. 

MnTMPyP rescues NRF2 protein synthesis at all inhibitory concentrations of H_2_O_2_ (Figure 2B). Interestingly, while MnTMPyP alone only slightly increases ARE reporter activity (Figure 2C, 1.0 ± 0.3 RU to 1.8 ± 0.1 RU), it increases reporter activation by 300 µM H_2_O_2_ more than 10-fold. These results suggest that H_2_O_2_ concentrations that allow NRF2 to escape from KEAP1 sequestration and activate the pathway maximally also generate other reactive species, perhaps hydroxyl radicals, which inhibit NRF2 protein synthesis and are blocked by MnTMPyP. MnTMPyP’s catalase activity may have also neutralized the H_2_O_2_ to an extent, but given the strong induction of the ARE reporter by the combination of MnTMPyP and H_2_O_2_, this is likely limited.

### 3.3. NRF2 Synthesis Is Stalled by H_2_O_2_-Induced Global Protein Synthesis Inhibition

We next examined if H_2_O_2_-mediated NRF2 synthesis suppression is caused by inhibition of global protein synthesis. Newly synthesized proteins were fluorescently labeled with OPP, a puromycin analog, to measure global translation activity. After 2 h, H_2_O_2_ had significantly reduced nascent global protein synthesis in a dose-dependent manner starting at 50 µM (Figure 3), similar to the dose-dependent effect of H_2_O_2_ on NRF2 synthesis (Figure 2B). Addition of MnTMPyP alone had no effect on global protein synthesis (0.97 ± 0.09 and 1.0 ± 0.1 relative pixel intensity with and without MnTMPyP, respectively). MnTMPyP with 25 µM H_2_O_2_ was slightly inhibitory compared to 25 µM H_2_O_2_ alone, indicating that at low H_2_O_2_ concentrations, the combination causes some cellular stress. However, MnTMPyP with 100 µM H_2_O_2_ significantly rescues the expression of newly synthesized proteins, closely mimicking the effect of MnTMPyP on NRF2 protein synthesis. The correlation between the effects of H_2_O_2_ ± MnTMPyP on NRF2 and global protein synthesis strongly suggests that NRF2 synthesis is stalled by H_2_O_2_-induced global protein synthesis inhibition.

We note that the high level of ARE reporter expression rescue by MnTMPyP in Figure 2C at 300 µM H_2_O_2_ could reflect some increased glutathionylation of KEAP1 under these conditions [36,37]. However, the rescue of both global protein synthesis (Figure 3) and NRF2 protein synthesis (Figure 2B) from H_2_O_2_ inhibition by MnTMPyP indicate that the observed rescue is, in large part, due to the relief of protein synthesis inhibition.

### 3.4. The ROS-Generating Molecule dtBHQ Also Inhibits NRF2 and Global Protein Synthesis; Opposing Roles for H_2_O_2_ in the NRF2/ARE Pathway

We next examined the effects of the hydroquinone dtBHQ, which generates H_2_O_2_ and other ROS by two-electron redox cycling (Figure 4A). A transition metal, e.g., Cu^2+^, catalyzes the sequential two-electron reduction of the hydroquinone, with electrons delivered to oxygen to generate superoxide. Reduction of the quinone by a cellular reductase creates a redox cycle. Unlike other oxidizable phenols, the quinone form of dtBHQ is prevented from acting as an electrophile due to the steric bulk of the two *tert*-butyl groups [38]. Previously [28], we found that, similar to H_2_O_2_, dtBHQ only weakly activates the NRF2/ARE pathway compared to sulforaphane in HaCaTs, shown by the ARE reporter assay and by expression of heme oxygenase-1 (HO-1) and aldo–keto reductase family 1 member C1 (AKR1C1), ARE-dependent cytoprotective proteins, and dtBHQ inhibits NRF2 protein synthesis in HaCaTs. At the same time however, H_2_O_2_ and dtBHQ significantly enhance sulforaphane’s ARE reporter activity [28], with dtBHQ acting synergistically [27]. DtBHQ causes fairly mild oxidative stress within 2 h, as treatment of HaCaT cells with up to 50 µM resulted in little change in total and oxidized glutathione levels [28]. First, we hypothesized that similar to the external H_2_O_2_ treatment, dtBHQ would suppress global protein synthesis, and thereby stall NRF2 synthesis, and that MnTMPyP would rescue both. DtBHQ (12.5 to 50 µM) inhibits NRF2 protein synthesis (Figure 4B), and MnTMPyP partly rescues this inhibition (Figure 4C). Global protein synthesis is indeed significantly suppressed by 50 µM dtBHQ and is largely rescued by MnTMPyP (Figure 4D and Appendix A, representative gel image). These data indicate that dtBHQ generates ROS that inhibit translation, and that MnTMPyP reverses this inhibition, restoring global protein synthesis and, with it, NRF2 synthesis.

Second, opposing inhibitory and stimulatory effects of H_2_O_2_ on the NRF2–ARE pathway are revealed in combination with sulforaphane. As dtBHQ and H_2_O_2_ inhibit NRF2 protein synthesis, they can also block NRF2 protein stabilization by an NRF2-activating electrophile such as sulforaphane. As shown in Figure 4E, NRF2 protein accumulation induced by 2.5 µM sulforaphane is substantially reduced by co-treatment with dtBHQ or H_2_O_2_. Global protein synthesis is also diminished by 2.5 µM sulforaphane + dtBHQ (Appendix A), while remaining unaffected by sulforaphane alone (Figure 4D). While this might, at first, suggest that activation of the NRF2/ARE pathway by sulforaphane would be inhibited by acute oxidative stress, e.g., co-treatment with dtBHQ or H_2_O_2_, we found previously that the opposite was true—the co-treatments significantly enhance both ARE reporter activation and expression of HO-1 and AKR1C1 [28]. Figure 4F (for dtBHQ) and Figure 4G (for H_2_O_2_) illustrate that despite low levels of total NRF2 present in the combination treatments (Figure 4E), the low levels of ARE activation by single treatments are greatly increased by the combination treatments. For example, 100 µM H_2_O_2_ enhances the ARE reporter activity of 2.5 µM sulforaphane from ~5-fold to ~35-fold. In sum, H_2_O_2_ has opposing effects on the pathway. It both reduces NRF2 protein levels by inhibiting global protein synthesis and activates the NRF2/ARE pathway downstream of KEAP1 cysteine oxidation/NRF2 accumulation.

## 4. Discussion

While mechanisms for increased translation of NRF2 protein in response to oxidative stress have been identified [16,17,18], these results show that ROS can inhibit rather than stimulate nascent NRF2 synthesis in human keratinocytes, by inhibiting global protein synthesis. This newly revealed mechanism of how the NRF2/ARE response can be paradoxically diminished by oxidative stress provides a potential explanation for other cell types that display a relatively weak ability of H_2_O_2_ to induce NRF2 protein accumulation and activation of ARE-driven gene expression, compared to electrophiles [14,23].

The question arises as to whether H_2_O_2_ itself inhibits NRF2 protein synthesis/global protein synthesis, or whether it is converted to a different reactive species that causes the inhibition. ROS/RNS are notoriously difficult to study in isolation in the cellular environment. H_2_O_2_ can readily react with the superoxide-generating hydroxyl radical or with nitric-oxide-generating peroxynitrite (Figure 2A), both of which are strong oxidizing agents, as opposed to H_2_O_2_, which is relatively slow and weak. MnTMPyP, but not catalase, rescued NRF2 protein levels in response to dtBHQ treatment, and MnTMPyP, but not catalase, can act to reduce hydroxyl radical and peroxynitrite levels. Catalase, which is highly specific for H_2_O_2_, instead inhibited ARE reporter activation by SFN +dtBHQ [28]. Thus, H_2_O_2_ likely enhances ARE-driven gene expression by targeting specific KEAP1 cysteines and acting on targets downstream of KEAP1 cysteines, while other species generated intracellularly from H_2_O_2_, e.g., the hydroxyl radical, inhibit NRF2/global protein translation.

A complete understanding of how H_2_O_2_ inhibits global protein synthesis is still emerging, driven in large part by studies in yeast [1]. While regulation can occur at any of the four stages of synthesis (initiation, elongation, termination, and ribosomal recycling), the initiation step appears to be a primary target of H_2_O_2_ and stressors in general. The integrated stress response (ISR) is the canonical mechanism by which stressors inhibit global protein synthesis; ER stress, amino acid deprivation, heme deprivation, and viral infection each activate one of four eIF2α kinases [2]. Phosphorylated eIF2α is a component of the eIF2 complex, which forms a ternary complex with GTP and Met-tRNAi, a requisite step in formation of the 43S pre-initiation complex, leading to the initiation of mRNA translation and recognition of the AUG start codon. Interestingly, oxidative stress has been shown to activate all eIF2α kinases in various studies, dependent in part on cell type [2,3,4,5]. In yeast, oxidative stress has been proposed to activate the ISR by a variety of mechanisms, e.g., oxidation of amino acids that cause amino acid starvation [6]. H_2_O_2_ can also inhibit the mTOR pathway, thereby attenuating both proper assembly of initiation factors to the mRNA cap (attenuating cap-dependent translation initiation) and translation elongation through elongation factor 2 (eEF2) [1]. Interestingly, Liu et al. showed in HEK293 and MEF cells that while ≥100 µM H_2_O_2_ inhibited translation through the mTOR pathway, eIF2α phosphorylation was significantly more sensitive to H_2_O_2_, responding to levels as low as 5 μM [7]. Additional means by which H_2_O_2_ regulates protein translation are likely to be found, as a proteome-wide investigation of protein thiols oxidized in H_2_O_2_-exposed yeast revealed a striking overrepresentation of ribosomal proteins and proteins with ribosome biogenesis and cytosolic translation functions [8].

ROS have been implicated in NRF2 suppression in several clinical models of disease, including diabetic kidney disease [39], Chagas disease [40], and Friedrich’s ataxia (FRDA) [41,42], with antioxidants rescuing the NRF2 response in these studies. For example, NRF2 protein accumulation in response to tBHQ was significantly attenuated in FRDA patient fibroblasts compared with control fibroblasts, and treatment with the antioxidant EUK-134 restored tBHQ and NRF2 responsiveness in patient cells. This apparently paradoxical effect of ROS-scavengers, enhancing rather than blocking NRF2 activation, reflects our observations in keratinocytes. Multiple possible mechanisms of NRF2 suppression by ROS have been identified [43], and for a given pathological condition, several of these may be involved. At least under acute oxidative stress conditions, the suppression of global protein synthesis can inherently limit NRF2 protein accumulation. It remains to be investigated whether suppression of global protein synthesis/NRF2 protein synthesis might be a contributing factor under chronic oxidative stress encountered in a particular disease state.

## 5. Conclusions

The NRF2/ARE response to stimuli requires de novo NRF2 protein synthesis due to its rapid degradation by the proteasome. To the best of our knowledge, this is the first study to measure NRF2 protein synthesis in response to a treatment that generates oxidative species. Inhibition of the proteasome allows NRF2 synthesis to be measured independently of any changes in the rate of its degradation. We find that concentrations of H_2_O_2_ sufficient to free NRF2 from KEAP1 repression and activate ARE-driven expression inhibit NRF2 protein synthesis in HaCaT keratinocytes. This appears to be a direct consequence of suppressed global protein synthesis, a well-known cellular response to oxidative stress. This mechanism may explain the relatively limited induction of the NRF2/ARE pathway by H_2_O_2_ as compared to electrophiles previously observed by others in MEFs and mouse keratinocytes [14,23].

However, several mechanisms have been identified by which NRF2 protein synthesis can be increased in response to H_2_O_2_ treatment [16,17,18]. Whether H_2_O_2_ or other oxidative species overall suppress or inhibit NRF2 protein synthesis in any given system likely depends on the cell type and cell culture conditions. 

## Figures and Tables

**Figure 1 antioxidants-12-01735-f001:**
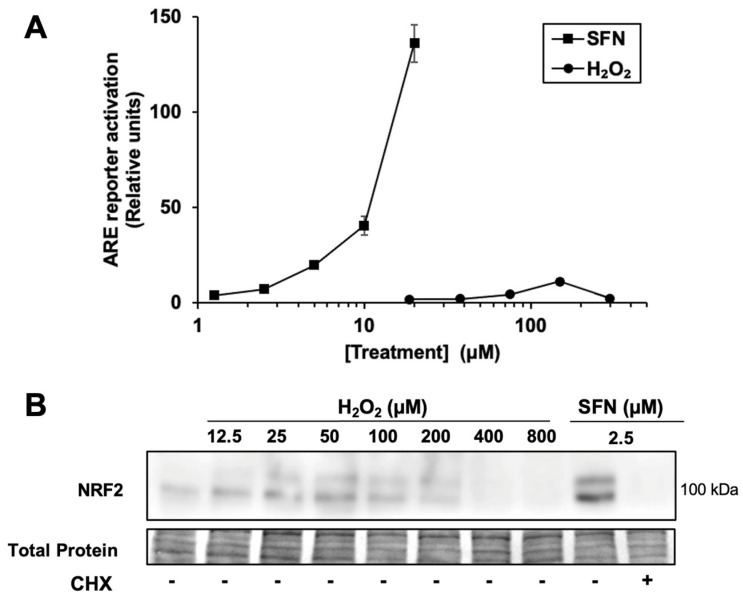
**Hydrogen peroxide is a relatively weak activator of the NRF2 pathway.** (**A**) Cells transfected with the ARE luciferase reporter and a control reporter were treated for 18 h with sulforaphane (SFN) or H_2_O_2_ at the concentrations shown. Luciferase reporter activity in lysates is ARE reporter activity divided by control reporter activity, *n* = 4, then normalized to vehicle-treated cells. Where error bars are not shown, they are shorter than the height of the symbol. (**B**) Cells were treated for 2 h with SFN or H_2_O_2_ prior to harvest and Western blotting. Where indicated, cycloheximide (CHX) was included with SFN. Total protein analysis was performed by Fast Green staining. Images are representative of two biological replicates with two technical replicates each.

**Figure 2 antioxidants-12-01735-f002:**
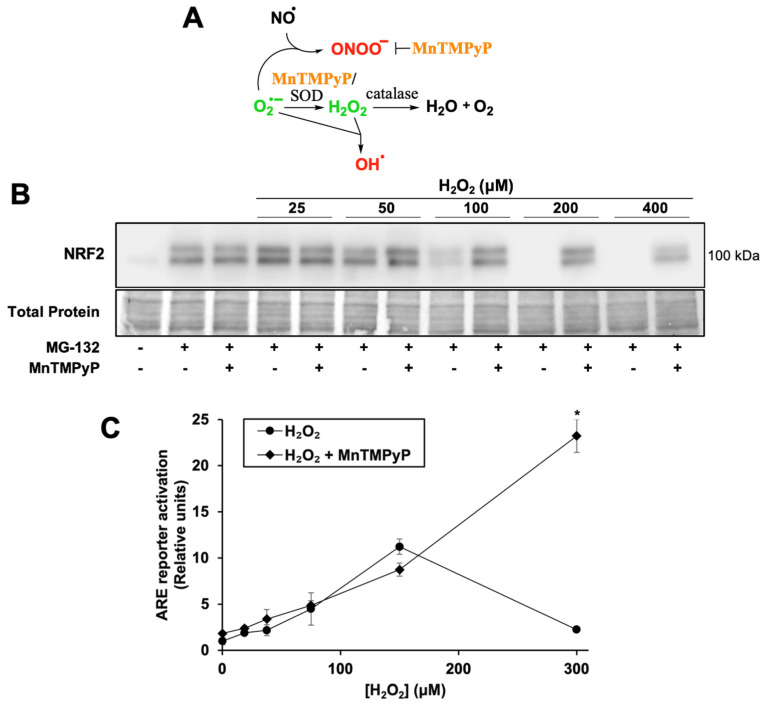
H_2_O_2_ treatment inhibits NRF2 protein synthesis; MnTMPyP rescues NRF2 protein synthesis and enhances ARE reporter activation. (**A**) Schematic of the antioxidant mechanism of MnTMPyP, see text for details. (**B**) Cells were treated for 2 h with SFN, H_2_O_2,_ and/or MnTMPyP. Proteasome inhibitor MG-132 was added 10 min post-treatment. Total protein assessed by Fast Green staining. Images are representative of four biological replicates with two technical replicates each. (**C**) Cells were treated for 18 h with MnTMPyP and/or H_2_O_2_ and processed as for Figure 1A. Where error bars are not shown, they are shorter than the height of the symbol. * *p* < 0.001 for H_2_O_2_ versus H_2_O_2_ + MnTMPyP. No asterisk indicates *p* > 0.05.

**Figure 3 antioxidants-12-01735-f003:**
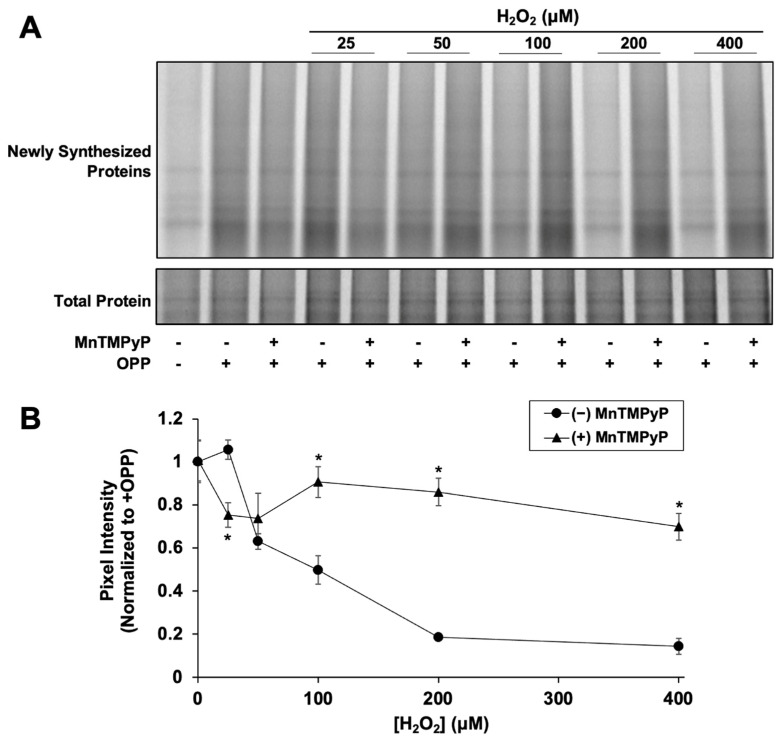
**H_2_O_2_ inhibits global protein synthesis.** Cells were treated for 80 min with H_2_O_2_ and MnTMPyP as indicated prior to addition of OPP or vehicle. Cells were incubated for an additional 40 min prior to harvest and global protein synthesis analysis. (**A**) Newly synthesized protein signal is AZdye 647 fluorescence. Total protein assessed by Coomassie staining. Images are representative of two biological replicates with two technical replicates each. (**B**) Quantitation of pixel intensity for all replicates, normalized to total protein per lane. * *p* < 0.001 for H_2_O_2_ versus H_2_O_2_ + MnTMPyP. No asterisk indicates *p* > 0.05.

**Figure 4 antioxidants-12-01735-f004:**
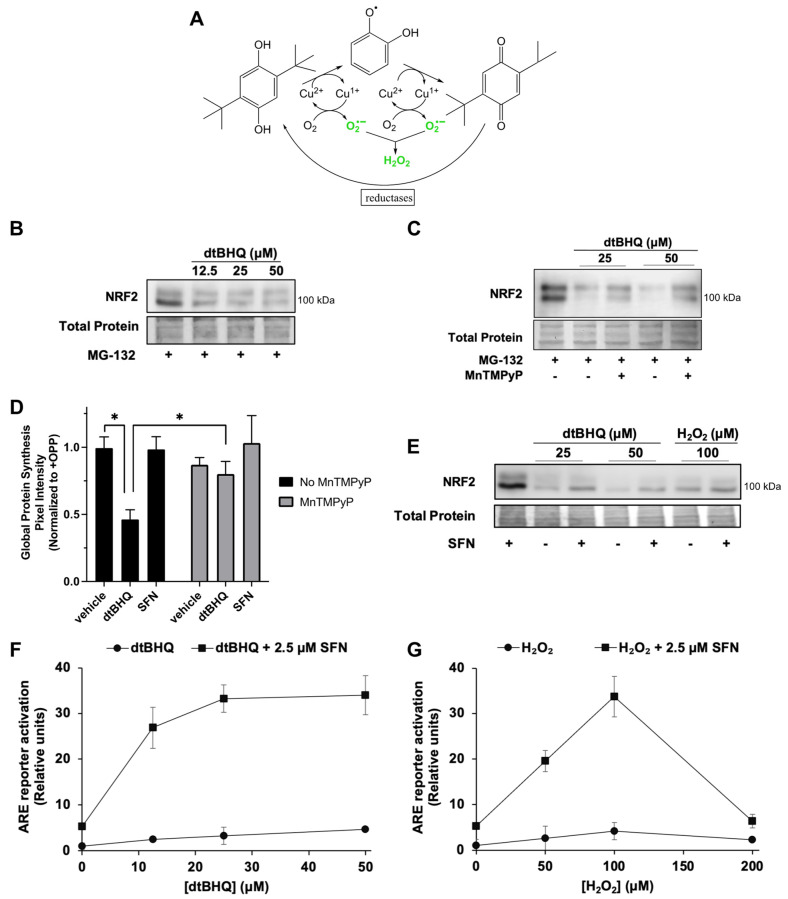
**The ROS-generating molecule dtBHQ inhibits NRF2 and global protein synthesis; opposing inhibitory and stimulatory roles for ROS/H_2_O_2_ in the NRF2/ARE pathway**. (**A**) Schematic of the generation of ROS by dtBHQ. See text for details. (**B**) Cells were treated for 2 h with dtBHQ prior to Western blot analysis. MG-132 was added 10 min post-treatment. (**C**) Treatments are as for B, with MnTMPyP added with dtBHQ as indicated. (**D**) Cells were treated for 2 h with 50 µM dtBHQ, 2.5 μM sulforaphane, and MnTMPyP as indicated, prior to harvest and global protein synthesis analysis as for Figure 3. * *p* < 0.001. (**E**) Cells were treated for 2 h with 2.5 μM sulforaphane, dtBHQ, and H_2_O_2_ as indicated, prior to Western blot analysis. (**F**,**G**) Cells were treated with 2.5 μM sulforaphane and/or H_2_O_2_ (**F**) or dtBHQ (**G**) 18 h prior to harvest as for Figure 1A. Where error bars are not shown, they are shorter than the height of the symbol. For Western blots (**B**,**C**,**E**), images are representative of two biological replicates with two technical replicates each, and total protein was assessed by Fast Green staining.

## Data Availability

The data presented in this study are available in the article/Appendix A; further inquiries can be directed to the corresponding author.

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
