# Peer review of "Acute Oxidative Stress Can Paradoxically Suppress Human NRF2 Protein Synthesis by Inhibiting Global Protein Translation"

_antioxidants, 2023, doi:10.3390/antiox12091735_

Round 1

Reviewer 1 Report

In this study, the authors found that H2O2 treatment could inhibit the Nrf2 protein synthesis and global protein synthesis in HaCaT keratinocytes. The results are reasonable. However, I don’t think these findings can lead to the conclusion that “the induction of the Nrf2/ARE pathway by H2O2 can be limited by its inhibition of Nrf2 protein synthesis’. It is well-known that treatment of high concertation of H2O2 could not only induce the oxidative stress, but also cause severe injury to the cell membrane and the cell contents. The results in this study can indicate that H2O2 could suppress human Nrf2 protein synthesis by inhibiting global protein translation. It cannot be stated that acute oxidative stress can paradoxically suppress human Nrf2 protein synthesis by inhibiting global protein translation. The title and the conclusion should be revised.

Author Response

Reviewer 1 Comments:

In this study, the authors found that H2O2 treatment could inhibit the NRF2 protein synthesis and global protein synthesis in HaCaT keratinocytes. The results are reasonable. However, I don’t think these findings can lead to the conclusion that “the induction of the NRF2/ARE pathway by H2O2 can be limited by its inhibition of NRF2 protein synthesis’. It is well-known that treatment of high concertation of H2O2 could not only induce the oxidative stress, but also cause severe injury to the cell membrane and the cell contents. The results in this study can indicate that H2O2 could suppress human NRF2 protein synthesis by inhibiting global protein translation. It cannot be stated that acute oxidative stress can paradoxically suppress human NRF2 protein synthesis by inhibiting global protein translation. The title and the conclusion should be revised.

Authors’ Responses:

We are grateful to the reviewer for the feedback on the manuscript, and we appreciate that the reviewer thought the study supported the conclusion that H2O2 suppresses human NRF2 protein synthesis by inhibiting global protein translation. The concern is that treatment of cells with a high concentration of H2O2 not only induces oxidative stress, but also causes injury to the cellular membrane and contents. Thus, changes to the title and conclusion are requested to instead state that H2O2 treatment, but not acute oxidative stress, can paradoxically suppress human NRF2 protein synthesis by inhibiting global protein translation.

First, we’d like to specifically address the concern that “high concertation of H2O2 could not only induce the oxidative stress, but also cause severe injury to the cell membrane and the cell contents“ with the following:

  1. Given that suppression of NRF2 protein synthesis and global protein synthesis by H2O2 and dtBHQ are all ameliorated by the antioxidant MnTMPyP, we think this strongly suggests that oxidative stress is responsible for the inhibition, rather than severe injury to the cell membrane and cell contents.
  2. We note that previously we found that dtBHQ, up to 50 µM, causes relatively low levels of oxidative stress within two hours, as total or oxidized glutathione levels showed little change in HaCaT cells [1]. We have added this to the manuscript, lines 598 to 600. Thus, oxidative species generated by dtBHQ, rather than severe injury to the cell, are likely responsible for protein synthesis inhibition.
  3. We note that H2O2 is commonly used in the NRF2 field as a representative oxidative insult in cell culture experiments, and the concentration of H2O2 in the manuscript that suppresses NRF2 protein synthesis and global protein synthesis (~100 µM) is at or below those typically used (300 and 600 µM, [2]; 130 and 400 µM, [3]; and 200 µM [4,5]). Similar ranges are used in assessing H2O2 effects on protein synthesis (please see the paragraph added to the discussion starting on line 712, requested by reviewer 3). Thus, while oxidation of the cell membrane and cellular components does occur, these concentrations of H2O2 have sub-lethal activities and are commonly used to deduce mechanisms of oxidative stress on NRF2 and protein synthesis.

In addition, regarding the requested change in the title and conclusion, we believe that “acute oxidative stress” is more accurate than “H2O2”, given the results with dtBHQ. DtBHQ is a redox cycling molecule that generates superoxide, which can be converted not only to hydrogen peroxide, but also to many other reactive oxygen or nitrogen species (e.g. Figure 2A). As discussed in the paragraph starting on line 700, the evidence from results with dtBHQ suggests that “H2O2 likely enhances ARE-driven gene expression by targeting specific KEAP1 cysteines and acting on targets downstream of KEAP1 cysteines, while other species generated intracellularly from H2O2, e.g. the hydroxyl radical, inhibit NRF2/global protein translation.” H2O2 treatment of cells will also result in a myriad of oxidized and oxidative species intracellularly as H2O2 reacts with cellular superoxide, or is converted in a Fenton reaction to the hydroxyl radical, causes oxidation of membrane lipids, etc. The term “oxidative stress” captures this cellular state, and “acute” refers to the bolus dose, rather than a chronic, sustained condition.

Finally, the thoughtful question raised by this reviewer, as to whether H2O2 or more generally, oxidative species/oxidative stress, are responsible for global protein synthesis suppression, prompted us to re-examine our graphical abstract and abstract. In the original submission, these stated that H2O2 was responsible for global protein synthesis inhibition. However, based on the points above, we changed the abstract and graphical abstract slightly to avoid specifically ascribing NRF2/GPS synthesis inhibition to H2O2.

References

  1. Bauman, B.M.; Jeong, C.; Savage, M.; Briker, A.L.; Janigian, N.G.; Nguyen, L.L.; Kemmerer, Z.A.; Eggler, A.L. Dr. Jekyll and Mr. Hyde: Oxidizable Phenol-Generated Reactive Oxygen Species Enhance Sulforaphane’s Antioxidant Response Element Activation, Even as They Suppress Nrf2 Protein Accumulation. Free Radic. Biol. Med. 2018, 124, 532–540, doi:10.1016/j.freeradbiomed.2018.06.039.
  2. Fernández-Ginés, R.; Encinar, J.A.; Hayes, J.D.; Oliva, B.; Rodríguez-Franco, M.I.; Rojo, A.I.; Cuadrado, A. An Inhibitor of Interaction between the Transcription Factor NRF2 and the E3 Ubiquitin Ligase Adapter β-TrCP Delivers Anti-Inflammatory Responses in Mouse Liver. Redox Biol. 2022, 55, 102396, doi:10.1016/j.redox.2022.102396.
  3. Suzuki, T.; Muramatsu, A.; Saito, R.; Iso, T.; Shibata, T.; Kuwata, K.; Kawaguchi, S.; Iwawaki, T.; Adachi, S.; Suda, H.; et al. Molecular Mechanism of Cellular Oxidative Stress Sensing by Keap1. Cell Rep. 2019, 28, 746-758.e4, doi:10.1016/j.celrep.2019.06.047.
  4. Fourquet, S.; Guerois, R.; Biard, D.; Toledano, M.B. Activation of Nrf2 by Nitrosative Agents and H2o2 Involves Keap1 Disulfide Formation. J Biol Chem 2010, 285, 8463–8471, doi:10.1074/jbc.M109.051714.
  5. Li, W.; Thakor, N.; Xu, E.Y.; Huang, Y.; Chen, C.; Yu, R.; Holcik, M.; Kong, A.-N. An Internal Ribosomal Entry Site Mediates Redox-Sensitive Translation of Nrf2. Nucleic Acids Res. 2010, 38, 778–788, doi:10.1093/nar/gkp1048.

Reviewer 2 Report

In the presented work the authors used keratinocytes cell line to verify if H2O2-dependent activation of NRF2 may inhibit NRF2-dependent protein synthesis. The authors conclude that induction of the Nrf2/ARE pathway by H2O2 can be limited by its inhibition of Nrf2 protein synthesis, likely by its arrest of global protein synthesis.

The aim of the study is clear and interesting, however the methodology and results do not support author’s conclusion.

First of all, the authors did not measure any protein which is known to be synthetized after NRF2/ARE activation like glutathione synthase, glutathione reductase, NQO1 etc.

Next, the authors stimulate cells for 2h or more with H2O2. For instance, it was found that a half-life of NRF2 is around 20 min. Why the authors choose to stimulate cells for 2h (e.g. fig. 1)? The inhibitory effect of higher doses of H2O2 may be caused by earlier activation of NRF2 (Kobayashi et al. Adv. Enzyme Regul. 46, 113–140 (2006).).

Also, I assume that the authors isolate whole cell lysate, not nuclear extract. Therefore, the authors analysed the NRF2 level in cytosol, where NRF2 is not transcriptionally active. Hence, basing on their results the authors cannot conclude that H2O2 may or may not activate NRF2. Additionally, the authors should show bands for control protein not for total protein content. When the authors add bands for control protein content (in proper cellular fraction) it will be possible to determine semi-quantitatively, the level of NRF2 activation (ex. see Piechota-Polanczyk et al. doi: 10.1155/2018/2028936, Fig 1d).

Lastly, the authors should also increase the number of biological replicates to at least 3. In fig. 3 the authors state that 2 biological replicates were done which is not enough to perform statistical analysis.

Reviewer 3 Report

This is a very interesting paper, which sheds new light on the mechanisms of ROS-induced NRF2 activation. It is well written and easy to follow. The methodology is correct and conlusions are supported by the results. 

I would also appreciate adding a paragraph in the discussion about possible mechanisms how ROS inhibit protein translation.

According to the correct nomenclature, protein names should be in capital letters (NRF2).

Author Response

Reviewer 3 Comments:

This is a very interesting paper, which sheds new light on the mechanisms of ROS-induced NRF2 activation. It is well written and easy to follow. The methodology is correct and conlusions are supported by the results. 

I would also appreciate adding a paragraph in the discussion about possible mechanisms how ROS inhibit protein translation.

According to the correct nomenclature, protein names should be in capital letters (NRF2).

Authors’ Responses:

We are appreciative of the reviewer’s time in reviewing the manuscript. We are delighted that it is of significant interest to the reviewer and that they find the methodology sound and the conclusions supported.

We are happy to add the requested paragraph to the discussion, which has been inserted at line 712.

We are grateful for the notification that the protein names should be in capital letters, and this has been corrected throughout.

Round 2

Reviewer 1 Report

The authors have adequately answered my concerns. However, in my opinion, the content in the response letter could be included in the introduction part to make this manuscript easy to understand. Meanwhile, the references, especially these in Introduction part, could be updated.

Author Response

We are grateful to the reviewer for their time and their suggestion to include an explanation in the Introduction, as to the use of H2O2 in the NRF2 field as a common means of inducing oxidative stress. This has been included in the first paragraph of the Introduction as shown with tracked changes. This addition definitely increases the overall clarity of the work. 

Three references were added with this explanation, and to our knowledge, the other references are updated. 

Reviewer 2 Report

The authors adressed all my questions and concerns. I am satisfied with their answer.

Author Response

We are grateful for the reviewer's time and contributions to the manuscript.